# Durability of ChAdOx1 nCoV-19 (Covishield^®^) Vaccine Induced Antibody Response in Health Care Workers

**DOI:** 10.3390/vaccines11010084

**Published:** 2022-12-30

**Authors:** Alka Verma, Amit Goel, Harshita Katiyar, Prachi Tiwari, Asari Sana, Dheeraj Khetan, Dharmendra Singh Bhadauria, Ajay Raja, Neelam Khokher, Ratendra Kumar Singh, Amita Aggarwal

**Affiliations:** 1Department of Emergency Medicine, Sanjay Gandhi Postgraduate Institute of Medical Sciences, Lucknow 226014, India; 2Department of Gastroenterology, Sanjay Gandhi Postgraduate Institute of Medical Sciences, Lucknow 226014, India; 3Department of Transfusion Medicine, Sanjay Gandhi Postgraduate Institute of Medical Sciences, Lucknow 226014, India; 4Department of Nephrology, Sanjay Gandhi Postgraduate Institute of Medical Sciences, Lucknow 226014, India; 5Additional Chief Medical Office, Directorate of Medical & Health Services, Lucknow 226014, India; 6Nursing Superintendent, Sanjay Gandhi Postgraduate Institute of Medical Sciences, Lucknow 226014, India; 7Department of Gastroenterology and Human Nutrition Unit, All Indian Institute of Medical Sciences, New Delhi 110029, India; 8Department of Clinical Immunology & Rheumatology, Sanjay Gandhi Postgraduate Institute of Medical Sciences, Lucknow 226014, India

**Keywords:** COVID-19, coronavirus, Covishield, COVID vaccine, Anti-SARS-CoV-2 antibody, neutralising antibody

## Abstract

(i) Background: ChAdOx1 nCoV-19 (Covishield^®^) vaccine is widely used in India. We studied the Covishield^®^ induced antibody response and its durability among health care workers (HCWs) (ii) Method: HCWs received two doses (0.5 mL) four weeks apart. Blood specimens, collected before each dose, day (D) 60, D150 and D270 after second dose, were tested for anti-spike antibody (ASAb) titre and neutralising antibody (%) (NAb) using Elecsys Anti-SARS-CoV-2 S (Roche) and SARS-CoV-2 NAb ELISA Kit (Invitrogen), respectively. Data are expressed as proportions and median (interquartile range) and compared using non-parametric (iii) Result: Among 135 HCWs (83 males; age 45 (37–53); 36 had pre-existing ASAb), 29 (21.5%) acquired COVID-19 after 60 (39–68) days of vaccination. ASAb titre before second dose and at D60, D150, D270 were 77.2 (19.4–329.4), 512 (114.5–9212), 149 (51.6–2283) and 2079 (433.9–8644) U/mL, respectively. Compared to those without pre-existing ASAb, titres were significantly higher before second dose (5929 vs. 41, *p* < 0.001), D60 (3395 vs. 234, *p* = 0.007) and D150 (1805 vs. 103, *p* < 0.001) in participants with pre-existing ASAb; NAb were also higher (80 vs. 18, *p* < 0.001) before second dose. Between those who acquired infection or not after vaccination, ASAb titres were comparable before second dose (77 vs. 78, *p* = 0.362) but significantly higher at D60 (14,019 vs. 317, *p* < 0.001) and D150 (2062 vs. 121, *p* = 0.002) in the former group, though NAb percentage were higher at D60 (87 vs. 27, *p* < 0.001) and D150 (79 vs. 25, *p* = 0.007) only (iv) Conclusions: Covishield^®^ induces a higher antibody titre in those with pre-existing ASAb. The vaccine induced antibody starts falling 5 months after vaccination.

## 1. Introduction

Severe acute respiratory syndrome coronavirus 2 (SARS-CoV-2), the etiological agent of COVID-19 illness, has caused over 0.6 billion virologically confirmed cases and over 6.5 million deaths [1] The pandemic has posed an unprecedent professional risk for health care workers (HCWs). Estimate suggests ~10% of the total burden of SARS-CoV-2 infection was borne by HCWs [2]. As compared to the community, HCWs had 11.6 times higher risk of testing positive for COVID-19 nucleic acid test [3]. Further, HCW had a 7.4 times higher risk of developing severe Covid as compared to non-essential workers [4]. According to the World Health Organization (WHO), 135 million HCWs were deployed between January 2020 and May 2021 to combat the pandemic and we lost 80,000–180,000 due to COVID-19 [5].

Coherent and joint efforts of the scientific community, internal health organizations, government agencies and the pharmaceutical industry led to the discovery of several vaccines against corona virus in a very short time period. After completion of necessary regulatory trials, front-line workers, including HCWs, were given corona vaccine as a priority. The ChAdOx1 nCoV-19 vaccine (AZD1222), developed at Oxford University, consists of a replication-deficient chimpanzee adenoviral vector ChAdOx1. It contain SARS-CoV-2 spike protein gene and has shown promising protective efficacy in large trials [6,7,8]. In India, this vaccine is manufactured by Serum Institute of India (Covishield^®^) and over 1.75 billion doses have been administered to date. The real-life data of Covishield^®^ shows promising results among HCWs, patients with liver cirrhosis [9], renal transplant recipients [10] and those on maintenance hemodialysis [11].

At present, data are limited on the persistence of protective antibodies in vaccine recipients. This study was conducted to study the serological immune response and temporal pattern of vaccine induced antibody response among HCWs who had received two doses of Covishield^®^ vaccine.

## 2. Methods

This prospective, observational, cohort study was conducted between January 2021 and November 2021 in the Covid vaccination facility of Sanjay Gandhi Postgraduate Institute of Medical Sciences, Lucknow, India. The HCWs of the institute, who were planned for vaccination were invited through electronic media (email, WhatsApp, text message) to participate in our study. Those who voluntarily agreed to participate were asked to contact our study center at the vaccination site. After obtaining written informed consent, they were requested to complete a data collection form followed by donation of their blood specimens. We enrolled all the participants from a single vaccination site. Participants were excluded if they were already vaccinated with any COVID-19 vaccine or had nucleic acid confirmed Covid infection in the preceding eight weeks. 

Two intramuscular 0.5 mL doses of Covishield^®^ were given four weeks apart. All the doses were administered in the vaccination facility established and manned by the state government. The vaccine dose and dosing interval were as dictated in the guidelines laid by the Government of India. The guidelines were in coherence with the contemporary literature [7]. Following each dose, recipients were observed for 30 min followed by telephonic contact for the next 48 h to report any significant adverse event. 

Specimen collection: We collected a 5.0 mL blood specimen at five-time points from each of the participants (a) within 48 h before administering the first dose, (b) within 48 h prior to the administration of second dose, (c) at 60 ± 7 days after second dose, (d) at 150 ± 14 days after second dose and (e) at 270 ± 14 days after second dose (Figure 1). Serum was separated, within one hour of its collection, by centrifugation at 4000× *g* for 10 min at 4 °C and stored in multiple aliquots at −80 °C temperature for serological testing at the end of the study. 

Serological testing: Sera were tested for anti-spike antibody (ASAb) titre and proportion of neutralising antibody (NAb) using Elecsys Anti-SARS-CoV-2 S (Roche Diagnostics, GmbH, Campus Vienna Biocenter 2, Vienna, Austria) and SARS-CoV-2 Neutralising Antibody ELISA Kit (Invitrogen, Catalog no BMS2326, Thermo-Fisher Scientific, Third Avenue, Waltham, MA, USA), respectively. Elecsys Anti-SARS-CoV-2 S immunoassay is an in vitro quantitative assay for antibodies against spike (S) protein receptor-binding domain (RBD) in humans. The test principle is based on an automated system’s double-antigen sandwich assay format. The assay had a sensitivity of 98.8% and specificity of 99.98%. The limit of quantitation for the assay is 0.40–250 U/mL. The specimens with antibody titre above 250 U/mL were serially diluted ×20, ×50 and ×100 fold to obtain results within the detection range. The titres that remained above the quantitation limit after 100-fold dilutions were reported as >25,000 U/mL. The antibody concentrations are expressed as U/mL and a value ≥ 0.80 U/mL is considered positive for the ASAb. The SARS-CoV-2 Neutralising Antibody ELISA Kit is a competitive ELISA assay. Specimens with calculated neutralisation ≥ 20% are considered positive. Both the assays were performed following the manufacturer’s recommendations. 

The trial data from India, available at the time of the start of the study suggested that 79% of people develop anti-Covid antibody after one dose [12]. We expected 75% response in our study, 10% relative error in the assumed response (75% ± 7.5%), two-sided 95% confidence interval, and estimated minimum required sample size was 129. After including expected loss of follow up, we targeted to include 150 HCWs. We could enroll and follow 135 participants.

Qualitative and quantitative data are expressed as numbers, proportions and median (interquartile range). Paired numerical data are compared using a non-parametric Wilcoxon signed-rank test. The data between the groups are compared using the Mann-Whitney test. The analyses were carried out using STATA 16 software. The level of significance was kept at <0.05. Our institute ethics committee approved the study (2021-16-IMP-EXP-118) and the participants were enrolled after obtaining written informed consent.

## 3. Results

We included data from 135 participants. Clinical characteristics of the participants are summarized in Table 1. Among the 17 participants, (12.6%) who had nucleic acid test confirmed prior COVID-19 infection 163 (145–174) days before the first dose, 14 (82.4%) had detectable ASAb and antibody titre was 82 (55–376) U/mL, and 10 of 17 (58.8%) tested positive for Nab (30.2 (13.6–40)). 

Of the 135 participants, 36 (26.7%) had detectable ASAb before vaccination and the titre was 114.5 (37.1–327) U/mL. Only 14 of 36 (38.9%) had symptomatic, nucleic acid test confirmed prior COVID-19 infection which indicate that 61.1% of infection remained asymptomatic. The ASAb titres of those with symptomatic (191.5 [70.3–520.6] U/mL) or asymptomatic (70.4 (7.2–279.3) U/mL) prior COVID-19 infection were comparable (*p* = 0.105). None of the 135 participants acquired COVID-19 infection between the administration of the two doses of vaccine but 29 (21.5%) acquired infection 60 (39–68) days after receiving the second dose. None of those who acquired infection after vaccination had serological evidence of prior covid infection. The vaccine effectiveness against delta variant related clinical illness was 79.5%. The ASAb titre and NAb percentage in the overall cohort are summarized in Table 2. The ASAb titre and proportion of NAb in log (natural) scale at various time points are depicted in Figure 2.

We compared the ASAb and NAb response of the participants with or without detectable ASAb before vaccination (Table 3). As compared to those without pre-existing ASAb, significantly higher ASAb titres were achieved in those with pre-existing ASAb at day 28 (*p* < 0.001), day 60 (*p* = 0.007) and day 150 (*p* < 0.001) but their levels were comparable at day 270 (*p* = 0.333) after second dose (Figure 3a). The NAb percentages were significantly higher at day 28 (*p* < 0.001) but were comparable at day 60 (*p* = 0.255), day 150 (*p* = 0.155) and day 270 (*p* = 0.626) (Figure 3b).

We compared the ASAb and NAb between those who acquired COVID-19 infection (*n* = 29) or not (*n* = 106) after complete vaccination (Table 4). Their ASAb titres were comparable after first dose (*p* = 0.362) but were significantly higher at day 60 (*p* < 0.001) and day 150 (*p* = 0.002) in those who acquired infection. The ASAb titres again became comparable at day 270 (*p* = 0.306). Similarly, their NAb percentages were comparable at day 28 (*p* = 0.071) but significantly higher at day 60 and 150 in those who developed infection. The NAb again became comparable between the groups at day 270 (*p* = 0.206). 

Among 99 participants without detectable ASAb before vaccination, 94 and 99 developed ASAb after first dose and second dose of vaccine and all had detectable ASAb in follow-up specimens collected at days 60, day 150 and day 270 after second doses (Table 5). The NAb was increased up to day 60 followed by a decrease in day 150 specimen. In follow-up at day 270, NAb proportion again increased from 54.5% to 73.4%. 

The anti-spike antibody titre value ≥ 0.80 U/mL is considered positive. The specimens with neutralisation ≥ 20% are considered positive. 

Following vaccination, 29 of 99 (29.3%) acquired Covid infection. On further analysis between those who acquired (*n* = 29) or remained protected (*n* = 70), the ASAb titre was comparable between the groups after the first dose of vaccine but the levels were significantly higher at day 60 and day 150 among those who developed infection after vaccination, though the titre became comparable between the groups at day 270 (Table 6).

## 4. Discussion

Frontline workers such as HCWs and security personnel received Covid vaccine as a priority in January 2021 in our country. In accordance with the contemporary scientific evidence, two 0.5 mL doses of ChAdOx1 nCoV-19 vaccine were administered at an interval of four weeks. All the HCWs who participated in our study had received the two doses by the end of February 2021. Soon after the complete vaccination of study participants, the second wave of the Covid pandemic due to the ‘Delta variant’ started in the country. 

We studied the temporal pattern of persistence of vaccine induced antibody up to 270 days after complete vaccination in 135 HCWs. We found that one fourth (26.7%) of HCWs had detectable ASAb before vaccination as evidence of prior COVID-19 infection, though only half had symptomatic Covid infection. Participants without pre-existing antibody developed detectable ASAb following a single dose of vaccine and these antibodies remained detectable at least for a period of nine months after complete vaccination. During the follow-up of nine months after complete vaccination, 21% of the subjects acquired symptomatic COVID-19 infection. This suggests a vaccine protection rate of 79%. We had not looked for asymptomatic Covid infection in vaccine recipients. In presence of pre-existing ASAb, the vaccine administration induced much higher antibody titres, though this booster effect waned after 9 months from the second dose of vaccine. In addition, data suggest that a natural Covid infection, following complete vaccination, further enhances the antibody response, which is sustained for a few months. 

Twenty percent of HCWs had detectable ASAb prior to vaccination suggesting prior infection. Several studies from India have shown high seroprevalence of antibodies to SARS-CoV-2 virus among HCW, though the seroprevalence varies from 2.5% in Kashmir [13] to 25% in New Delhi [14]. In India, the seroprevalence were higher in studies conducted in a sparsely populated hilly area [13,15] as compared to that reported from densely populated metropolitan cities [14,16,17]. High seroprevalence among HCWs is likely to be related to their inherent risk for repeated exposure and high-level of exposure during patient care. 

Among those with pre-existing antibodies before vaccination, only half had symptomatic COVID-19 infection. A recent systematic review of 350 studies from across the world have estimated that 35.1% (95% CI: 30.7 to 39.9%) of infections were asymptomatic [EN_419]. A slightly higher proportion of asymptomatic infection in our study might be due to the fact that HCWs enrolled in our study might have ignored mild illness because of their professional responsibilities, according to the need of the hour everywhere. 

Following vaccination, nearly all the participants seroconverted, similar to in other studies, conducted in HCWs, which reported a seroconversion rate of 99.5% [18] and 97.1% in the United Kingdom [19]. The high rates of seroconversion seen in our study might have been influenced by occurrence of asymptomatic natural infection following vaccination, because a second wave of corona virus started soon after the participants received their second dose of vaccine. 

Second dose boosted the immune response by more than 6 fold (from 77 U/mL to 512 U/mL). The original AZD1222 vaccine in trial also showed a booster effect from the second dose [7]. Non-inferiority of Covishield^®^ over original AZD1222 vaccine is proven in a good quality, double-blinded, randomized controlled trial [20]. This study also showed a 4 fold rise in ASAb (9786 AU to 38,576 AU) titre after second dose of vaccine. We have previously reported a 36-fold rise in antibody titre after two doses in immunosuppressed renal transplant recipients [10]. As compared to those who did not have ASAb (Table 3), the booster effect in participants with pre-existing ASAb showed peculiar features of marked booster effect after first dose without additional immune boosting after second dose. This suggests that the booster effect of vaccine is likely to be more intense in the presence of natural infection induced immunity as compared to vaccine induced immunity. Probably, in the presence of prior natural infection, a single dose of vaccine may be sufficient to achieve protective effect. Similar booster effects in the presence of prior natural infection have been reported by other groups [21,22]. Similarly, a natural infection had a booster effect even after the completion of a two dose schedule. Among those without evidence of prior natural infection, the ASAb titres were much higher on day 60, 150 and 270 in those who acquired natural infection as compared those who remained protected (Table 6). 

The ASAb titre peaked at day 60 followed by rapidly cline. The serum ASAb titres were markedly reduced at day 150 (Table 2). Though, data on durability of vaccine induced antibodies are limited [23], the decline in natural infection induced antibody titre with time is reported [24]. In people on anticancer therapy, the ChAdOx1-nCoV-19 vaccine-induced antibody is shown to start decline soon after the second dose [23]. A large population-based cohort study from Brazil (*n* = 42,558,839) and Scotland (*n* = 1,972,454) showed that, after completion of a two dose schedule of ChAdOx1-nCoV-19 vaccine, its effectiveness against both severe COVID-19 and symptomatic infection diminished over time. This decline in effectiveness starts as soon as 5–7 weeks after the second dose. Such a decline in vaccine effectiveness is a surrogate marker of declining vaccine induced immunity [25]. We observed a rise in ASAb titre at day 270. This could be related to subclinical infection during the ongoing delta wave between day 150 and day 270. The delta wave was waning at that time but HCWs were still in contact with patients. 

The vaccine effectiveness against delta variant to prevent clinical illness in overall cohort was 79.5%. All those who acquired covid infection after vaccination had no evidence of either clinical or serological prior covid infection. Among those who did not have ASAb before vaccination, the vaccine effectiveness was 70.3%. Combined data from four large randomized trials, which used Astra Zeneca vaccine, showed a vaccine efficacy of 62.1% [8]. The studies from India showed that complete vaccination with Covishield^®^ yield an efficacy of up to 88.6% [26]. Other studies from India have also shown that ChAdOx1-nCoV-19 vaccine led to reduced severity of illness [27] as well as risk of mortality during the second wave of the pandemic [28].

Our study had a few strengths. All the participants were recruited and vaccinated within a very short period. The blood specimens, from all participants, were also collected over a period of 2–3 days which reduced the chance for variable risk of natural infection among the participants. We studied the temporal pattern of vaccine-induced antibody response. We had data on pre-vaccination ASAb status which helped us to study the effect of vaccine on those with or without prior covid infection. Our participants, during and after vaccination, were actively involved in providing services to infected patients, which consistently posed a threat for breakthrough infection; hence the clinical effectiveness reflected in our data is an underestimation of vaccine effectiveness in the community. 

The limitations of our study include lack of an unvaccinated control group, which limit the analysis of effectiveness of the vaccine; lack of genomic sequencing of the variant of the coronavirus which caused break through infection is another important limitation of our study.

Several questions remain unanswered and their answers will be needed in future. We will need to follow our cohort to know the long-term durability, antibody titre, neutralising capacity and effectiveness of vaccine-induced antibodies in either preventing new infections or reducing the severity of clinical illness following infection with novel variants, which could appear in future. Further, we also need to study the cell mediated immunity achieved with vaccination. Further data will be needed to study the effect of dosing interval on the durability of immune response.

## 5. Conclusions

ChAdOx1 nCoV-19 (Covishield^®^) vaccine induces detectable ASAb and NAb in all recipients after two doses. The antibody titre induced by vaccination is much higher in those who have had prior covid infection. The vaccine induced antibody persists for at least two months and starts falling 5 months after second dose.

## Figures and Tables

**Figure 1 vaccines-11-00084-f001:**
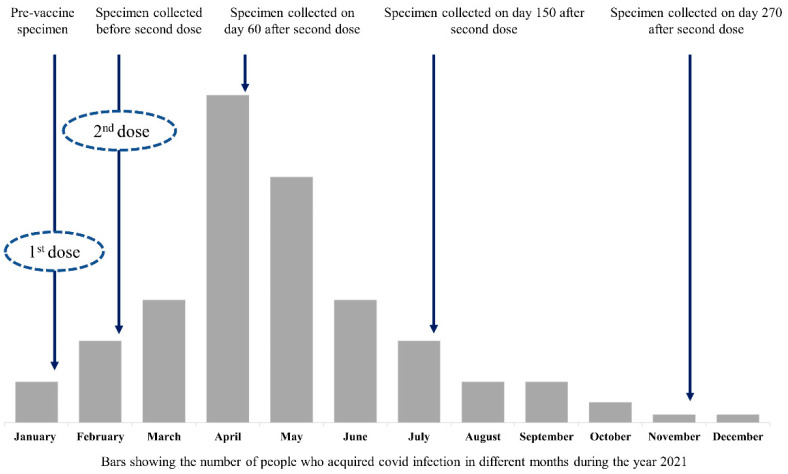
Temporal pattern of the Covid pandemic in the city, time of vaccination and time of blood specimen collection over the period of study.

**Figure 2 vaccines-11-00084-f002:**
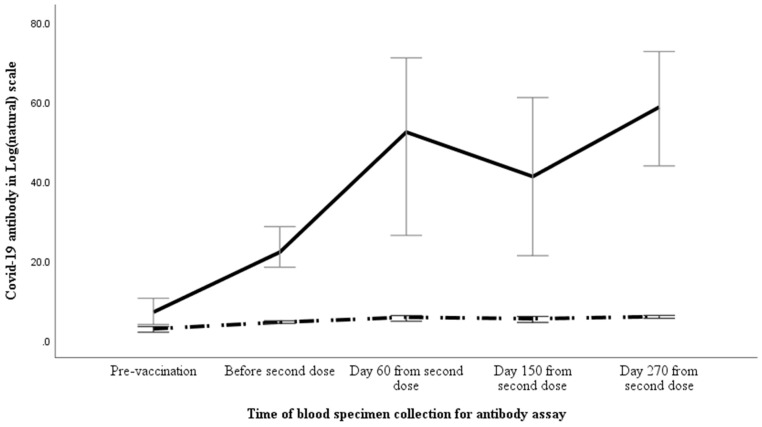
Temporal pattern of anti-spike antibody (solid line) and percentage of neutralising antibody (broken line) in study participants. The ASAb and NAb results along with 95% error bar are shown in Log (natural) scale along the *Y* axis, whereas *X* axis shows the time of specimen collection.

**Figure 3 vaccines-11-00084-f003:**
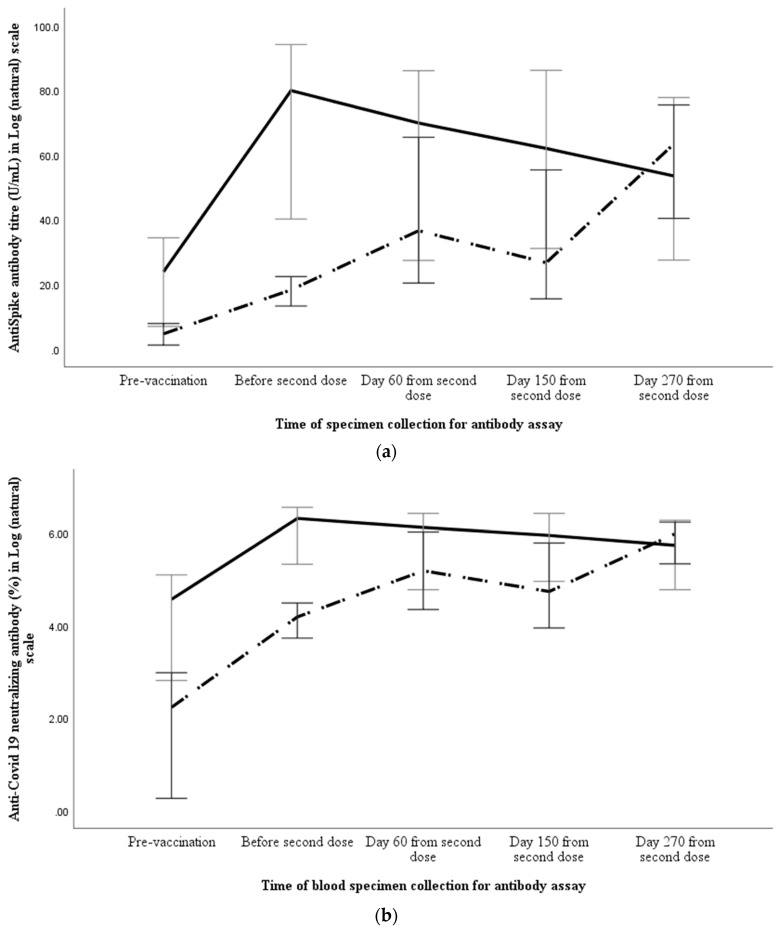
Comparison of anti-spike antibody titre (**a**) and percentage of neutralising antibody (**b**) between those with (solid line) or without (broken line) detectable anti-spike antibody before vaccination. The anti-spike antibody and neutralising antibody results, along with 95% error bar, are shown in Log (natural) scale along the *Y* axis, whereas *X* axis shows the time of specimen collection.

**Table 1 vaccines-11-00084-t001:** Characteristics of the study participants (*n* = 135).

Variable	Value
Males	83 (61.5)
Age (years)	45 (37–53)
Associated conditions	
Hypertension	15 (11)
Diabetes mellitus (DM)	7 (5)
Hypothyroidism	5 (4)
Bronchial asthma	3 (2)
Coronary artery disease	2 (2)
Rheumatoid arthritis	2 (2)
Osteoarthritis	1 (1)
Paroxysmal supraventricular tachycardia	1 (1)
Bronchial asthma + hypertension	1 (1)
Total	37 (27)
Body mass index (Kg/M^2^)	25.2 (22.9–27.6)
Prior COVID-19 infection	17 (13)

Categorical data are presented as number and proportions; numerical data are expressed as median (interquartile range).

**Table 2 vaccines-11-00084-t002:** Temporal pattern of vaccine induced COVID-19 antibody in overall cohort (*n* = 135).

COVID-19 Antibody	Time of Specimen Collection (Days)			
	After the First Dose	60 Days after Second Dose	150 Days after Second Dose	270 Days after Second Dose
Anti-Spike antibody	77.2 (19.4–329.4)	512 (114.5–9212)	149 (51.6–2283)	2079 (433.9–8644)
Neutralising antibody (%)	22.1 (8.5–54.2)	52.3 (7.7–92.7)	41.1 (3.8–86.4)	58.6 (17.6–85.1)

Anti-spike antibody titre is expressed as U/mL and neutralising antibody is expressed in percentage. The anti-spike antibody titre value ≥ 0.80 U/mL is considered positive. The specimens with calculated neutralisation ≥ 20% are considered positive for neutralising antibody. Data are expressed as median (inter quartile range).

**Table 3 vaccines-11-00084-t003:** Comparison of COVID-19 antibody response between those with or without pre-existing anti-spike antibody before vaccination in overall cohort (*n* = 135).

Anti-COVID Antibody	COVID-19 Antibody	Time of Specimen Collection (Days)							
		Before Second Dose at 28 Days		60 Days after Second Dose		150 Days after Second Dose		270 Days after Second Dose	
Anti-Spike antibody (U/mL)	Anti-spike antibody was absent before vaccination (*n* = 99)	40.9 (11.8–118.3)	*p* < 0.001	234.3 (96.3–9268)	*p* = 0.007	102.6 (41.4–1837)	*p* < 0.001	1858 (215.8–6784)	*p* = 0.333
	Anti-spike antibody was present before vaccination (*n* = 36)	5929.5 (649.4–16,207.5)		3395 (800.1–8852)		1805 (201.9–4131)		2423.5 (707.1–9552)	
Neutralising antibody (%)	Anti-spike antibody was absent before vaccination (*n* = 99)	18.3 (5.9–32)	*p* < 0.001	36.6 (7.4–91.9)	*p* = 0.255	26.7 (3.6–85.5)	*p* = 0.155	63.4 (17.6–85.4)	*p* = 0.626
	Anti-spike antibody was present before vaccination (*n* = 36)	80 (24.1–95.8)		69.9 (12.6–94.2)		62.1 (13.2–89)		53.6 (17–84.1)	

Anti-spike antibody titre is expressed as U/mL and neutralising antibody is expressed in percentage. The anti-spike antibody titre value ≥ 0.80 U/mL is considered positive. The specimens with calculated neutralisation ≥ 20% are considered positive for neutralising antibody. Data are expressed as median (interquartile range).

**Table 4 vaccines-11-00084-t004:** Comparison of COVID-19 antibody response between the participants who acquired COVID-19 infection (*n* = 29) or not (*n* = 106) after the second dose of vaccine (*n* = 135) in overall cohort.

COVID-19 Antibody	Time of Specimen Collection (Days)							
	Before the Second Dose		60 Days after Second Dose		150 Days after Second Dose		270 Days after Second Dose	
	Acquired COVID-19 Infection	Not Acquired COVID-19 Infection	Acquired COVID-19 Infection	Not Acquired COVID-19 Infection	Acquired COVID-19 Infection	Not Acquired COVID-19 Infection	Acquired COVID-19 Infection	Not Acquired COVID-19 Infection
Anti-Spike antibody (U/mL)	76.8 (23.4–200.4)	78.3 (19.3–684.9)	14,019 (4546–25,000)	316.7 (102.3–3447)	2062 (139–4317)	120.7 (50.9–1584)	2185 (1414–4611)	2052 (215.8–10,177)
*p* value	0.362		<0.001		0.002		0.306	
Neutralising antibody (%)	18 (7.6–27.5)	24.5 (8.7–66.6)	86.9 (66.6–97)	27.3 (5.7–83.1)	79.3 (45–90.8)	24.6 (3.3–85)	67.7 (54–84.7)	50.3 (12.6–85.2)
*p* value	0.071		<0.001		0.007		0.206	

Anti-spike antibody titre is expressed as U/mL and neutralising antibody is expressed in percentage. The anti-spike antibody titre value ≥ 0.80 U/mL is considered positive. The specimens with calculated neutralisation ≥ 20% are considered positive for neutralising antibody. Data are expressed as median (inter quartile range).

**Table 5 vaccines-11-00084-t005:** Proportion of participants, without anti-spike antibody before vaccination, who developed COVID-19 antibody after vaccination (*n* = 99).

COVID-19 Antibody	Time of Specimen Collection (Days)			
	After the First Dose	60 Days after Second Dose	150 Days after Second Dose	270 Days after Second Dose
Anti-Spike antibody (%)	94 (95)	99 (100)	99 (100)	99 (100)
Neutralising antibody (%)	46 (46.5)	61 (61.6)	54 (54.5)	73 (73.4)

**Table 6 vaccines-11-00084-t006:** Comparison of COVID-19 antibody response between those who acquired COVID-19 infection (*n* = 29) or not (*n* = 70) after the second dose of vaccine in the cohort of those without detectable anti-spike antibody before vaccination (*n* = 99).

COVID-19 Antibody	Time of Specimen Collection (Days)							
	Before the Second Dose		60 Days after Second Dose		150 Days after Second Dose		270 Days after Second Dose	
	Acquired COVID-19 Infection	Not Acquired COVID-19 Infection	Acquired COVID-19 Infection	Not Acquired COVID-19 Infection	Acquired COVID-19 Infection	Not Acquired COVID-19 Infection	Acquired COVID-19 Infection	Not Acquired COVID-19 Infection
Anti-Spike antibody (U/mL)	76.8 (23.4–200.4)	36.2 (10.2–98.6)	14,019 (4546–25,000)	147.5 (85.4–387.7)	2062 (139–4317)	64.3 (37.4–154.3)	2185 (1414–4611)	1450 (128–12,885)
*p* value	0.052		<0.001		<0.001		0.147	
Neutralising antibody (%)	18 (7.6–27.5)	18.4 (5.8–32.9)	86.9 (66.6–97)	21 (5.3–65.5)	79.3 (45–90.8)	16.1 (2.8–64.8)	67.7 (54–84.7)	48.7 (12.6–85.8)
*p* value	0.936		<0.001		0.001		0.277	

Anti-spike antibody titre is expressed as U/mL and neutralising antibody is expressed as a percentage. The anti-spike antibody titre value ≥ 0.80 U/mL is considered positive. The specimens with calculated neutralisation ≥ 20% are considered positive for neutralising antibody. Data are expressed as median (interquartile range).

## Data Availability

The detailed data are available upon reasonable request to the corresponding author.

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
