# Peer review of "Durability of ChAdOx1 nCoV-19 (Covishield®) Vaccine Induced Antibody Response in Health Care Workers"

_vaccines, 2022, doi:10.3390/vaccines11010084_

Round 1

Reviewer 1 Report

How the doses are selected? Please refer to the reference (two doses (0.5 ml)

How did you determine the interval between each dose? Please refer to the reference (HCWs received two doses (0.5 ml) four weeks apart)

It is better for the authors to expand future aspects are missing.

How was the sample size selected? Please mention the sample formula

It should also be mentioned the sampling method and where and what center the patients were selected from

The “p” of p-value must be written in italics and small letter. Some words must be written in italics and small letter.- Please add Town, State, and Nation when citing pharmaceutical or device industries.

It is better for the authors to revise and expand their discussion-Mention each of your important findings, then compare with others and mention your interpretation and inference of the results and causes of similarities and differences.

The p values are given in the all of tables

It is better for the authors to carefully edited by a fluent native English speaker

Author Response

We have provided our response in a separate sheet

Reviewer 2 Report

Please edit the paper for legibility. 

For example:

Line 61.  'Till date, over 2.2 billion 61 doses have been administered to date.'

could be improved avoiding date x 2.

Figs 2 is confusing. Where is the scale for the broken line?

Thanks

Author Response

We have provide dour respons ein a separate sheet
